# Physical activity levels in adults and elderly from triaxial and uniaxial accelerometry. The Tromsø Study

Edvard H. Sagelv[1]*, Ulf Ekelund[2,3], Sigurd Pedersen[1], Søren Brage[4,5], Bjørge H. Hansen[6], Jonas Johansson[7], Sameline Grimsgaard[7], Anna Nordström[1,8], Alexander Horsch[9], Laila A. Hopstock[7], Bente Morseth[1]

**1** School of Sport Sciences, Faculty of Health Sciences, UiT the Arctic University of Norway, Tromsø, Norway, **2** Department of Sports Medicine, Norwegian School of Sport Sciences, Oslo, Norway, **3** Department of Chronic Diseases and Ageing, the Norwegian Institute for Public Health, Oslo, Norway, **4** MRC Epidemiology Unit, University of Cambridge, Cambridge, United Kingdom, **5** Department of Sports Science and Clinical Biomechanics, Faculty of Health Sciences, Southern Denmark University, Odense, Denmark, **6** Department of Sport Science and Physical Education, Faculty of Health Sciences, University of Agder, Agder, Norway, **7** Department of Community Medicine, Faculty of Health Sciences, UiT the Arctic University of Norway, Tromsø, Norway, **8** Department of Public Health and Clinical Medicine, Umeå University, Umeå, Sweden, **9** Department of Computer Science, Faculty of Natural Sciences, UiT the Arctic University of Norway, Tromsø, Norway

* edvard.h.sagelv@uit.no

## Abstract

### Introduction

Surveillance of physical activity at the population level increases the knowledge on levels and trends of physical activity, which may support public health initiatives to promote physical activity. Physical activity assessed by accelerometry is challenged by varying data processing procedures, which influences the outcome. We aimed to describe the levels and prevalence estimates of physical activity, and to examine how triaxial and uniaxial accelerometry data influences these estimates, in a large population-based cohort of Norwegian adults.

### Methods

This cross-sectional study included 5918 women and men aged 40–84 years who participated in the seventh wave of the Tromsø Study (2015–16). The participants wore an ActiGraph wGT3X-BT accelerometer attached to the hip for 24 hours per day over seven consecutive days. Accelerometry variables were expressed as volume (counts·minute$^{-1}$ and steps·day$^{-1}$) and as minutes per day in sedentary, light physical activity and moderate and vigorous physical activity (MVPA).

### Results

From triaxial accelerometry data, 22% (95% confidence interval (CI): 21–23%) of the participants fulfilled the current global recommendations for physical activity ($\geq$150 minutes of MVPA per week in $\geq$10-minute bouts), while 70% (95% CI: 69–71%) accumulated $\geq$150

**Data Availability Statement:** Data availability: The legal restriction on data availability are set by the Tromsø Study Data and Publication Committee in order to control for data sharing, including

publication of datasets with the potential of reverse identification of de-identified sensitive participant information. The data can however be made available from the Tromsø Study upon application to the Tromsø Study Data and Publication Committee. Contact information: The Tromsø Study, Department of Community Medicine, Faculty of Health Sciences, UiT The Arctic University of Norway; e-mail: tromsous@uit.no.

**Funding:** The article processing charges are funded by the publication fund at the University Library at UiT the Arctic University of Norway. The work of Søren Brage was funded by the UK Medical Research Council [MC_UU_12015/3] and the NIHR Biomedical Research Centre in Cambridge [IS-BRC-1215-20014].

**Competing interests:** The authors declare no competing interests.

minutes of non-bouted MVPA per week. When analysing uniaxial data, 18% fulfilled the current recommendations (i.e. 20% difference compared with triaxial data), and 55% (95% CI: 53–56%) accumulated ≥150 minutes of non-bouted MVPA per week. We observed approximately 100 less minutes of sedentary time and 90 minutes more of light physical activity from triaxial data compared with uniaxial data (p<0.001).

## Conclusion

The prevalence estimates of sufficiently active adults and elderly are more than three times higher (22% vs. 70%) when comparing triaxial bouted and non-bouted MVPA. Physical activity estimates are highly dependent on accelerometry data processing criteria and on different definitions of physical activity recommendations, which may influence prevalence estimates and tracking of physical activity patterns over time.

## Introduction

Physical inactivity is the fourth-leading cause for premature mortality globally, and the health benefits of physical activity are undisputable [1–3]. Thus, surveillance of physical activity at the population level is crucial in order to track levels and trends of physical activity, which may support public health initiatives to promote physical activity [4].

Traditionally, physical activity is assessed using self-report methods, which are susceptible to recall and social desirability bias [5]. Over the last two decades, the use of objective approaches to measure bodily movements, such as accelerometers, have progressively increased and may complement self-reported measures in large scale population-based studies [6–9]. However, accelerometry measured physical activity levels vary across different populations, socioeconomic status, sex and body composition [10–15]. Although these differences may be true, inherent variations in device technology and data processing procedures influence the outcome [7] and may hamper the comparability between studies.

Additionally, more recent accelerometers measure acceleration in three axes (vertical, coronal and sagittal) [7], whereas older models that are used in many observational studies measured acceleration in the axial (vertical) plane only [6]. Triaxial accelerometers are expected to record a wider range of movement and activities than uniaxial accelerometers [16]. In laboratory studies, measures of standardized activities from uniaxial and triaxial accelerometry differs in adolescents [17], but are similar in adults [18]. However, in free-living studies of adults, triaxial accelerometry data detected more minutes in higher intensity physical activity [8] and a larger volume of sporting activities than uniaxial accelerometry data [19]. To our knowledge, no study has compared triaxial and uniaxial accelerometry data from the GT3X ActiGraph accelerometer in a large population-based sample during free-living conditions. Thus, considering the potential differences in triaxial and uniaxial data, comparisons of prevalence estimates in a large population sample are warranted.

The current global recommendations for physical activity suggests at least 150 minutes of moderate and vigorous physical activity (MVPA) per week in at least 10-minutes bouts [20]. Recently, new recommendations in the United States have omitted the bout length requirement [21]. When comparing prevalence estimates of bouted and non-bouted MVPA from uniaxial accelerometry, the proportions fulfilling the recommendations vary largely (1%-70%)

[10, 22, 23]. Although similar discrepancies may be expected from triaxial accelerometry, the proportional differences are unknown.

The aim of this study was to describe the levels and prevalence of physical activity in a large population-based cohort stratified by age, sex, body mass index (BMI) and educational level; and to compare potential differences in these estimates between triaxial and uniaxial accelerometry data.

## Materials and methods

### Design

The Tromsø Study is an ongoing population-based cohort study in the municipality of Tromsø, Norway. The study invites participants from previous surveys as well as random samples in repeated surveys (Tromsø 1: 1974, Tromsø 2: 1979–80, Tromsø 3: 1986–7, Tromsø 4: 1994–95, Tromsø 5: 2001, Tromsø 6: 2007–08, Tromsø 7: 2015–16). The data collection consists of questionnaires and interviews, biological sampling and clinical examinations. The detailed design of the Tromsø Study is described elsewhere [24]. The present study includes participants from the seventh survey conducted in 2015–16.

In Tromsø 7, all inhabitants of Tromsø municipality aged 40 years and older (N = 32591) were invited to the first visit, of which 21083 (65%) attended. Of all invited participants to Tromsø 7, a sub-sample was invited back for a second visit that included more extensive examinations. This sub-sample (n = 13304) included 20% of the inhabitants 40–59 years (n = 4,008) and 50% of the inhabitants 60–84 years (n = 6,142) randomly drawn from the total sample. In addition, previous participants in selected clinical examinations in Tromsø 6 not already included in the random sample were added (n = 3,154). Of the 8346 attending the second visit, due to logistical reasons, 6778 were invited to wear an accelerometer, of which 6333 (93%) accepted. Participants without valid accelerometry data due to lost accelerometers (n = 6), returned accelerometers with technical error (n = 37) or with invalid wear time data (n = 165) were excluded. Accordingly, 6125 participants provided valid wear time of four days of at least 10 hours. Of these, 167 and 65 participants did not report their educational level and smoking habits, respectively, and 24 did not undergo weight and/or height measurement. With some failing to report two or more potential covariates, we ended up with a sample of 5918 participants aged 40–84 years with valid data on accelerometry measured physical activity and potential confounders, which are included in our analyses.

All participants gave written informed consent. Tromsø 7 and this present study were approved by the Regional Ethics Committee for Medical Research (REC North ref. 2014/940 and 2016/758410, respectively) and the Norwegian Data Protection Authority.

### Data collection

Height and weight were measured in light clothing without shoes. BMI was calculated as weight divided by the square of height (kg·m$^{-2}$) and defined as normal- and underweight ($<$25 kg·m$^{-2}$), overweight (25–29.9 kg·m$^{-2}$) or obese ($\geq$30 kg·m$^{-2}$), respectively. Information on educational level was collected from questionnaires and categorized in four groups; 1) primary school, 2) high school diploma, 3) University education $<$4 years and 4) University education $\geq$4 years.

Physical activity and sedentary behaviour were measured with an ActiGraph wGT3X-BT accelerometer (ActiGraph, LLC, Pensacola, United States), firmware versions 1.2.0- to 1.8.0. Trained technicians instructed each participant on how to wear the accelerometer before attaching the accelerometer to their right hip using an elastic band. Participants were instructed to wear the accelerometer for 24 hours a day for eight consecutive days and nights

(the rest of the day following the visit in the clinic and seven more days), perform their daily activities as usual, and only to remove the accelerometer during water-based activities (e.g. showering or swimming) and contact sports. The participants returned the accelerometer by mail in a pre-paid envelope. The ActiLife software (ActiGraph, LLC, Pensacola, United States) was used for initialisation and downloading the data. The accelerometer was initialized for raw data mode with a sampling frequency of 100 hertz and were set to start recording at 00:00 the day following the visit in the clinic.

### Accelerometry data processing

When reducing the raw acceleration files to epochs, the normal (default) filter in the ActiLife software was applied, which is proprietary to the manufacturer [7, 25]. The epochs were aggregated to 10 seconds. The .agd-files (epoch files) were further converted to .csv-files using the ActiLife software, which were thereafter analysed using the Quality Control & Analysis Tool (QCAT), a custom-made software for processing of accelerometry data developed in Matlab (The MathWorks, Inc., Natick, Massachusetts, USA). The acceleration units are expressed in triaxial vector magnitude (VM) (the square root of the sum of squared activity counts) counts per minute (CPM)), and as uniaxial CPM for data from the axial plane (vertical axis) only. The step count of the accelerometer was derived from the axial plane, based on a proprietary algorithm developed by the manufacturer.

The 10-second epoch data was summed to 1 minute, where each minute was classified as wear time if either its value was ≥5 VM CPM and there were at least 2 minutes ≥5 VM CPM on the proceeding or following 20-minute time span, or if its value did not exceed 5 VM CPM, but both on the preceding, and on the following 20-minute, there were 2 or more minutes of ≥5 VM CPM. Otherwise the acceleration was considered to be noise and classified as non-wear time [26]. Recordings containing at least four days with a minimum of 10 hours wear time each, were included in the analyses [7, 27]. All files flagged with invalid wear time data were visually inspected to confirm that the participants did not have valid wear time data (≤10 hours and ≤4 days). By visual inspection of diagrams from 30 random participants, the non-wear time algorithm appears to exclude sleep, which is thus defined as non-wear time in our analyses.

The triaxial VM CPM cut-points for different intensities were determined according to Peterson et al. [28] for sedentary behaviour and Sasaki et al. [29] for MVPA as follows: sedentary behaviour: <150, light physical activity: 150–2689, and MVPA: ≥2690 VM CPM. Intensity-specific cut-points for the axial plane were <100 CPM for sedentary behaviour, a cut-point originally determined for adolescents girls [30] but also later adopted for adults [31]. For light physical activity and MVPA, the uniaxial CPM cut-points were set between 100 and 1951 CPM and ≥1952 CPM, respectively [32]. The study by Peterson et al. [28] suggest that 100 uniaxial CPM are equivalent to 150 triaxial VM CPM. The studies by Sasaki et al. [29] and Freedson et al. [32] validated the respective cut-points using similar protocols that are matched in locomotion speeds on the treadmill and the movements should thus be biomechanically equivalent, resulting in comparable triaxial and uniaxial intensity specific cut-points for walking and running.

The following variables were extracted for our analyses: days of wear time, mean wear time per valid day of wear time, mean uniaxial CPM, mean triaxial VM CPM, mean steps per day, time (min·day$^{-1}$) spent in sedentary-, light-, moderate and vigorous intensity physical activity, and the percentage meeting the World Health Organisation (WHO)'s recommended levels of physical activity (i.e. ≥150 min of MVPA per week in ≥10-minute bouts) [20]. Participants who accumulated ≥22 mean minutes of MVPA per day in at least 10-minute bouts (i.e. 150

minutes per week divided by seven days) were considered meeting the recommendations. This criteria of 150 min of MVPA per week was also assessed in accumulated non-bouted MVPA [21]. We assumed that triaxial VM CPM would capture more movements than uniaxial CPM. Thus, physical activity estimates are primarily derived from triaxial VM CPM, which are compared to uniaxial CPM.

## Availability of data and materials

The full variable list for accelerometry estimates of physical activity data in the Tromsø Study is available at NESSTAR WebView tool [33]. The data that support the findings of this study are available from the Tromsø Study but restrictions apply to the availability of these data, which were used under license for the current study, and so are not publicly available. The data can however be made available from the Tromsø Study upon application to the Data and Publication Committee of the Tromsø Study [34]. The Matlab code for the QCAT software for the current study can be made available upon reasonable request to the corresponding author, however, the accelerometry data processing of epoch data was carried out in the QCAT software as described above. The QCAT software is under development and is planned to be made publicly available as an open source software in the future.

## Statistical analysis

All data were confirmed to be normally distributed by visual inspection of the residuals when performing univariate analyses of covariance (ANCOVA) to assess associations between physical activity measures and age (10-year age groups), sex, BMI and educational level, with mutual adjustment for each other (e.g. when analysing physical activity by BMI, these analyses are adjusted for sex, age, and education etc.) in addition to adjustment for smoking and height. Paired samples t-tests was performed to check for differences between triaxial and uniaxial results, without adjustments for covariates. Independent sample t-tests was performed to assess for differences in age, weight, height and BMI between the total sample and the accelerometer sample, in addition to assess for sex differences in descriptive variables, in both the total and the accelerometer sample. Finally, we performed Pearson´s chi square tests to assess differences in the distribution of BMI groups, educational level and smoking habits among those who were invited but declined to wear an accelerometer and those who were invited and accepted the invitation. The descriptive physical activity estimates are presented as unadjusted mean ± standard deviation (SD) unless otherwise is stated. The Statistical Package for Social Sciences (Version 25, International Business Machines Corporation, United States) was used to perform all statistical analysis.

## Results

Overall and sex specific participant characteristics of the total Tromsø 7 sample with valid data on covariates (BMI, education and smoking, N = 20485) are presented in Table 1. Overall and sex specific participant characteristics of the accelerometry sample (N = 5918) are presented in Table 2. There were no differences in BMI between the total sample and the accelerometry sample (p = 0.054), while age, height and weight differed between the total sample and the accelerometry sample (p<0.001). In the accelerometry sample, women had lower BMI, height and weight than men (all p<0.001). Age distribution varied, where the age group 60–69 years consisted of 42% of the sample. The majority of the sample was overweight, as 45.3% (n = 2681) and 22.6% (n = 1337) were classified as overweight and obese, respectively.

**Table 1. Participant characteristics.** The Tromsø Study total sample 2015–16.

| | Women | Men | Total |
|---|---|---|---|
| N | 10753 (52.5%) | 9732 (47.5%) | 20485 |
| Age (years) | 57.0 ± 11.3 | 57.2 ± 11.2 | 57.1 ± 11.3 |
| Height (cm)* | 164.3 ± 6.5 | 177.8 ± 6.7 | 170.7 ± 9.4 |
| Weight (kg)* | 72.6 ± 13.9 | 88.1 ± 14.2 | 80.0 ± 16.0 |
| BMI (kg·m$^{-2}$)* | 26.9 ± 4.9 | 27.8 ± 4.0 | 27.3 ± 4.5 |
| <25 | 4329 (64.9%) | 2337 (35.1%) | 6666 (32.5%) |
| 25–29.9 | 3997 (44.6%) | 4958 (55.4%) | 8955 (43.7%) |
| >30 | 2427 (49.9%) | 2437 (50.1%) | 4864 (23.7%) |
| Educational level | | | |
| Primary school | 2567 (54.4%) | 2149 (45.6%) | 4716 (23%) |
| High school | 2735 (48.0%) | 2963(52.0%) | 5698 (27.8%) |
| University <4 yrs | 1897 (47.8%) | 2070 (52.2%) | 3967 (19.4%) |
| University ≥4 yrs | 3554 (58.2%) | 2550 (41.8%) | 6104 (29.8%) |
| Smoking | | | |
| Daily | 1558 (54.8%) | 1288 (45.2%) | 2849 (13.9%) |
| Previous | 4706 (52.0%) | 4340 (48.0%) | 9046 (44.2%) |
| Never | 4489 (52.2%) | 4104 (47.8%) | 8593 (41.9%) |

BMI = body mass index. Data are shown as mean ± standard deviation or n (%). The presented relative (%) prevalence is horizontal between women and men, while in the total column vertical between groups of BMI, educational level and smoking. *Significant difference between women and men (p<0.001)

## Overall physical activity levels

On average, the participants wore the accelerometer for 6.8 (SD: 0.5) days, and 58 (1%), 151 (3%), 860 (15%) and 4849 (81%) participants provided four, five, six and seven days of ≥10 hours of wear time, respectively. Mean wear time per day was 17.3 (SD: 1.8) hours. The

**Table 2. Participant characteristics.** The Tromsø Study accelerometry sample 2015–16.

| | Women | Men | Total |
|---|---|---|---|
| N | 3172 (53.6%) | 2746 (46.4%) | 5918 |
| Age (years) | 63.4 ± 10.2 | 63.4 ± 10.1 | 63.3 ± 10.2 |
| Height (cm)* | 163.6 ± 6.3 | 176.9 ± 6.7 | 169.8 ± 9.3 |
| Weight (kg)* | 71.7 ± 12.9 | 86.9 ± 13.7 | 78.8 ± 15.3 |
| BMI (kg·m$^{-2}$)* | 26.8 ± 4.7 | 27.8 ± 3.9 | 27.2 ± 4.4 |
| <25 | 1218 (64.1%) | 682 (35.9%) | 1900 (32.1%) |
| 25–29.9 | 1270 (47.4%) | 1411 (52.6%) | 2681 (45.3%) |
| >30 | 684 (51.2%) | 653 (48.8%) | 1337 (22.6%) |
| Educational level | | | |
| Primary school | 1008 (58.2%) | 724 (41.8%) | 1732 (29.2%) |
| High school | 838 (50.1%) | 834 (49.9%) | 1672 (28.3%) |
| University <4 yrs | 515 (46.4%) | 594 (53.6%) | 1109 (18.7%) |
| University ≥4 yrs | 811 (57.7%) | 594 (42.3%) | 1405 (23.7%) |
| Smoking | | | |
| Daily | 396 (56.4%) | 306 (43.6%) | 702 (12%) |
| Previous | 1498 (47%) | 1405 (51%) | 2903 (49%) |
| Never | 1278 (40%) | 1035 (37%) | 2313 (39%) |

BMI = body mass index. Data are shown as mean ± SD or n (%). The presented relative (%) prevalence is horizontal between women and men, while in the total column vertical between groups of BMI, educational level and smoking. *Significant difference between women and men (p<0.001)

**Table 3. Volume measures and intensity specific minutes per day by sex.** The Tromsø Study accelerometry sample 2015–16.

| | Women (n = 3172) | Men (n = 2746) | Total (n = 5918) |
|---|---|---|---|
| Wear time per day (hr) | 17.2±1.7 | 17.3±1.9 | 17.3±1.8 |
| Uniaxial counts per minute | 249.4±103.9* | 264.5±119.9 | 256.4±111.87 |
| Vector magnitude counts per minute | 539.5±168.5 | 530.4±187.3 | 535.3±177.5 |
| Steps per day | 6999.9±2940.1 | 6932.7±2924.5 | 6968.7±2932.8 |
| Sedentary behaviour uniaxial (min·day$^{-1}$) | 687.8±93.7 | 704.8±104.5 | 695.7±99.2 |
| Sedentary behaviour triaxial (min·day$^{-1}$) | 574.4±94.2 | 604.7±103.4 | 588.5±99.7 |
| Light physical activity uniaxial (min·day$^{-1}$) | 318.2±78.3 | 300.2±81.6 | 309.9±80.4 |
| Light physical activity triaxial (min·day$^{-1}$) | 417.5±86.1* | 384.2±86.9 | 402.0±88.1 |
| MVPA uniaxial | | | |
| With 10-min bouts (min·day$^{-1}$) | 11.2±14.9 | 11.6±16.2 | 11.3±15.5 |
| Without 10-min bouts (min·day$^{-1}$) | 28.0±22.3* | 31.8±25.7 | 29.8±24.0 |
| MVPA triaxial | | | |
| With 10-min bouts (min·day$^{-1}$) | 13.2±16.2 | 13.7±18.3 | 13.4±17.2 |
| Without 10-min bouts (min·day$^{-1}$) | 38.4±27.6* | 44.0±32.3 | 41.0±30.0 |

Data are shown as unadjusted mean ± SD. The presented $P_{equality}$ derives from the ANCOVA and is adjusted for educational level, body mass index, height, age and smoking. MVPA = moderate and vigorous physical activity. *significant difference between women and men ($p < 0.05$).

participants accumulated a mean of 535 (SD: 2.3) VM CPM and 6968.7 (SD: 2932.8) steps per day. From triaxial accelerometry data, time spent in sedentary behaviour and light physical activity was 9.8 (SD: 1.7) and 6.7 (SD: 1.5) hours per day, respectively. The participants accumulated 41 (SD: 30) and 13 (SD: 17.2) minutes per day of non-bouted MVPA and bouted MVPA, respectively (Table 3).

## Physical activity levels by age, sex, BMI and educational level

There were no sex differences in volume estimates (VM CPM and steps per day) or in time spent sedentary (Table 3). Women accumulated more minutes of light physical activity than men ($p < 0.001$) and men accumulated more minutes of non-bouted MVPA than women ($p < 0.001$), while women and men accumulated an equal amount of bouted MVPA ($p = 0.08$) (Table 3). In total, 22% (95% C.I.: 21–23%) fulfilled the recommended levels of physical activity (determined as ≥22 minutes MVPA per day in ≥10-minute bouts), compared with 70% (95% CI: 69–71%) in accumulated non-bouted MVPA (Fig 1).

All physical activity measures were inversely associated with age ($p < 0.001$), except for time spent in sedentary behaviour ($p = 0.01$) (Table 4).

Steps per day and VM CPM were inversely associated with BMI ($p < 0.001$) (Table 5). Sedentary time was positively associated with BMI ($p = 0.02$), while light physical activity, accumulated non-bouted MVPA and bouted MVPA were inversely associated with BMI ($p < 0.001$) (Table 5).

Finally, VM CPM, steps per day and sedentary behaviour were not associated with educational level ($p > 0.06$). There were differences in light physical activity between educational levels ($p = 0.003$), and bouted MVPA were positively associated with educational level ($p = 0.02$). There were no differences in accumulated non-bouted MVPA between educational levels ($p = 59$) (Table 6).

## Triaxial versus uniaxial data processing

There were differences between all triaxial and uniaxial accelerometry estimates of physical activity (all $p < 0.05$) (Table 3, 4, 5 and 6). Data from triaxial accelerometry data resulted in

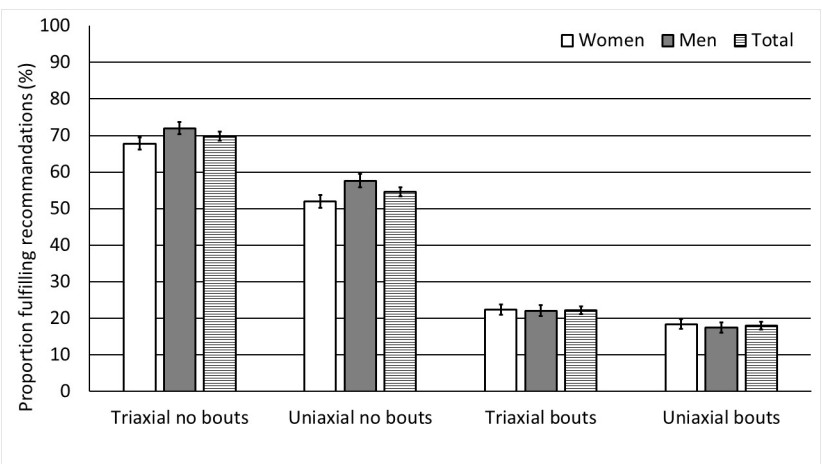

**Fig 1. The proportion of women (n = 3172) and men (n = 2746) separately, and in total (n = 5918), fulfilling the WHO´s recommendations for physical activity of 150 minutes of MVPA per week, in both accumulated non-bouted and bouted MVPA and from triaxial and uniaxial data.** Data is shown as percentage and error bars are 95% C.I.

~110 less minutes spent sedentary and ~90 more minutes spent in light physical activity compared with data from uniaxial accelerometry (p<0.001). A larger proportion of participants (22%, 95% C.I.: 21–23%) fulfilled the current physical activity recommendations when using triaxial data compared with analyses from uniaxial accelerometry (18%, 95% C.I.: 17–19%). For accumulated non-bouted MVPA, the corresponding prevalence estimates were 70% (95% C.I.: 69–71%) and 55% (95% C.I.: 53–56%) from tri- and uniaxial accelerometry, respectively (Fig 1).

Additionally, comparisons of tri- and uniaxial accelerometry resulted in different associations with age, sex, BMI and education; Women accumulated more minutes in light intensity physical activity than men from triaxial data (p<0.001), which was not observed from uniaxial data (p = 0.10) (Table 3). Sedentary time was positively associated with BMI from triaxial data (p = 0.02), but not from uniaxial data (p = 0.06) (Table 5). There was a difference in light physical activity between BMI groups from triaxial data (p<0.001), but not from uniaxial data (p = 0.06) (Table 5).

## Dropout analysis

There were no differences in distribution of smoking habits (p = 0.45) and BMI groups (p = 0.62) between participants who accepted and participants who declined the invitation to wear an accelerometer. A larger proportion of women than men declined the invitation to wear an accelerometer (p = 0.04), and participants who declined were older and had lower education than those who accepted the invitation (p<0.001).

## Discussion

In this population-based study of Norwegian adults and elderly, 22% fulfilled the current global recommendation for physical activity, however, when counting all accumulated non-bouted MVPA, the proportion increased three-fold, to 70%. Physical activity levels were inversely associated with older age and men accumulated more minutes of non-bouted MVPA than women. Those with lower BMI and higher education accumulated more minutes in MVPA. Furthermore, our results suggest higher prevalence estimates of sufficiently active

**Table 4. Volume measures and intensity specific minutes per day by age groups.** The Tromsø Study accelerometry sample 2015–16.

| | 40–49 years (n = 759) | 50–59 years (n = 986) | 60–69 years (n = 2501) | 70–79 years (n = 1437) | ≥80 years (n = 235) | P_equality |
|---|---|---|---|---|---|---|
| Wear time per day (hr) | 17.4±1.5 | 17.6±1.6 | 17.4±1.7 | 16.8±1.9 | 16.2±2.2 | <0.001 |
| Uniaxial counts per minute | 301.8±117.3 | 289.5±106.3 | 261.7±107.3 | 214.6±101.9 | 170.6±88.6 | <0.001 |
| Vector magnitude counts per minute | 609.3±179.3 | 578.9±166.6 | 542.5±172.4 | 475.5±167.4 | 402.1±142.6 | <0.001 |
| Steps per day | 8135.4±2814.0 | 7964.6±2756.8 | 7198.7±2831.5 | 5681.4±2631.6 | 4449.9±2448.7 | <0.001 |
| Sedentary behaviour uniaxial (min·day$^{-1}$) | 686.3±95.3 | 699.0±95.5 | 698.0±99.6 | 694.4±100.5 | 695.8±112.4 | 0.009 |
| Sedentary behaviour triaxial (min·day$^{-1}$) | 579.5±96.1 | 593.3±96.0 | 593.0±99.5 | 584.5±101.8 | 573.3±111.4 | 0.01 |
| Light physical activity uniaxial (min·day$^{-1}$) | 322.5 ±75.3 | 320.3±75.7 | 315.5±79.4 | 294.1±82.7 | 262.0±80.0 | <0.001 |
| Light physical activity triaxial (min·day$^{-1}$) | 409.8±83.3 | 408.4±83.6 | 405.7±87.3 | 391.6±93.3 | 376.7±87.0 | <0.001 |
| MVPA uniaxial | | | | | | |
| With 10 min bouts (min·day$^{-1}$) | 12.6±15.1 | 13.8±15.7 | 12.3±16.2 | 8.1±14.2 | 5.4±11.8 | <0.001 |
| Without 10 min bouts (min·day$^{-1}$) | 37.1±24.0 | 36.6±23.4 | 31.1±24.1 | 21.4±21.5 | 14.0±18.4 | <0.001 |
| MVPA triaxial | | | | | | |
| With 10 min bouts (min·day$^{-1}$) | 15.1±16.5 | 16.1±17.0 | 14.5±18.0 | 10.0±16.1 | 6.5±13.0 | <0.001 |
| Without 10 min bouts (min·day$^{-1}$) | 52.7±29.1 | 49.5±28.7 | 42.7±29.8 | 29.9±27.3 | 18.4±22.0 | <0.001 |

Data are shown as unadjusted mean ± SD. The presented P_equality derives from the ANCOVA and is adjusted for body mass index, sex, educational level, smoking and height. MVPA = moderate and vigorous physical activity.

participants from triaxial accelerometry data than from uniaxial accelerometry data, and we observed differences in all measures from tri- and uniaxial data, which was consistent across age, sex, BMI, and education.

Our prevalence estimates of physical activity based on accelerometry suggest that 1 out of 5 are fulfilling the current recommendations of ≥150 minutes per week of MVPA, which is

**Table 5. Volume measures and intensity specific minutes per day by BMI.** The Tromsø Study accelerometry sample 2015–16.

| | Normal weight (n = 1900) | Overweight (n = 2681) | Obese (n = 1337) | P_equality |
|---|---|---|---|---|
| Wear time per day (hr) | 17.5±1.7 | 17.2±1.8 | 17.0±1.9 | <0.001 |
| Uniaxial counts per minute | 279.7±119.2 | 256.6±109.7 | 222.8±95.9 | <0.001 |
| Vector magnitude counts per minute | 579.1±183.0 | 533.5±171.9 | 472.9±162.6 | <0.001 |
| Steps per day | 7857.7±3132.5 | 6929.1±2768.9 | 5784.7±2497.5 | <0.001 |
| Sedentary behaviour uniaxial (min· day$^{-1}$) | 698.2±101.4 | 692.4±96.4 | 699.0±100.7 | 0.06 |
| Sedentary behaviour triaxial (min· day$^{-1}$) | 575.4±101.3 | 587.3±96.4 | 609.3±100.7 | 0.02 |
| Light physical activity uniaxial (min· day$^{-1}$) | 314.7±81.0 | 312.0±79.8 | 298.7±79.7 | 0.06 |
| Light physical activity triaxial (min· day$^{-1}$) | 422.1±87.1 | 402.0±85.1 | 373.6±87.5 | <0.001 |
| MVPA uniaxial | | | | |
| With 10-min bouts (min· day$^{-1}$) | 15.6±18.0 | 10.8±14.8 | 6.2±10.9 | <0.001 |
| Without 10-min bouts (min· day$^{-1}$) | 35.7±25.6 | 29.5±23.4 | 21.9±20.2 | <0.001 |
| MVPA triaxial | | | | |
| With 10-min bouts (min· day$^{-1}$) | 17.8±19.4 | 13.1±16.7 | 7.9±12.8 | <0.001 |
| Without 10-min bouts (min· day$^{-1}$) | 47.0±31.4 | 40.8±29.6 | 32.9±26.7 | <0.001 |

Data are shown as unadjusted mean ± SD. The presented P_equality derives from the ANCOVA and is adjusted for age, sex, educational level, smoking and height. BMI = body mass index, MVPA = moderate and vigorous physical activity.

**Table 6. Volume measures and intensity specific minutes per day by education.** The Tromsø Study accelerometry sample 2015–16.

| | Primary School (n = 1732) | High School (n = 1672) | University <4 years (n = 1109) | University ≥4 years (n = 1405) | P$_{equality}$ |
|---|---|---|---|---|---|
| Wear time per day (hours) | 17.0±1.9 | 17.3±1.8 | 17.3±1.9 | 17.4±1.7 | 0.26 |
| Uniaxial counts per minute | 230.2±107.1 | 251.2±108.8 | 264.9±107.6 | 288.1±115.9 | 0.18 |
| Vector magnitude counts per minute | 505.4±178.5 | 533.3±178.7 | 538.6±171.9 | 571.9±172.5 | 0.58 |
| Steps per day | 6128.4±2803.5 | 6906.1±2819.9 | 7154.9±2828.9 | 7931.5±2991.6 | 0.07 |
| Sedentary behaviour uniaxial (min·day$^{-1}$) | 686.6±101.2 | 695.7±98.5 | 701.8±102.1 | 702.1±94.3 | 0.06 |
| Sedentary behaviour triaxial (min·day$^{-1}$) | 578.9±100.2 | 588.3±100.8 | 596.9±102.0 | 593.9±95.1 | 0.10 |
| Light physical activity uniaxial (min·day$^{-1}$) | 311.3±85.8 | 316.4±81.4 | 304.7±76.5 | 304.4±74.3 | 0.002 |
| Light physical activity triaxial (min·day$^{-1}$) | 404.9±94.0 | 407.8±87.5 | 394.3±85.6 | 397.8±82.2 | 0.003 |
| MVPA uniaxial | | | | | |
| With 10-min bouts (min·day$^{-1}$) | 7.9±13.5 | 9.8±13.9 | 12.4±15.1 | 16.5±18.3 | 0.02 |
| Without 10-min bouts (min·day$^{-1}$) | 23.1±22.4 | 28.1±22.7 | 32.0±22.7 | 38.2±25.6 | 0.06 |
| MVPA triaxial | | | | | |
| With 10-min bouts (min·day$^{-1}$) | 9.6±15.6 | 11.9±15.6 | 14.7±16.6 | 18.9±19.7 | 0.02 |
| Without 10-min bouts (min·day$^{-1}$) | 33.8±29.8 | 40.2±29.8 | 43.1±28.3 | 49.3±29.6 | 0.59 |

Data are shown as unadjusted mean ± SD. The presented P$_{equality}$ derives from the ANCOVA and is adjusted for sex, age, body mass index, smoking and height.

MVPA = moderate and vigorous physical activity.

substantially lower than the global estimate from self-reported physical activity in western high-income countries (~63%) [35]. As self-reported physical activity is prone to recall and social desirability bias, self-report may overestimate the true physical activity level [36], which may indicate that more accurate estimates can be derived from device-based assessments (e.g. accelerometry) [37]. Thus, understanding how different measurements tools may influence the prevalence estimates is important to inform public health recommendations and policies.

The WHO´s physical activity recommendations for health are primarily based on self-reported physical activity [20]. Recently, based on data from both self-report and accelerometry, the revised United States recommendations for physical activity omitted the requirement that MVPA should be performed in at least 10-minute bouts [21]. Although the domain or type of MVPA is unknown, non-bouted MVPA may represent more sporadic activities and small bursts of movements, which may include transportation, stair climbing or house work, compared to bouted MVPA, which may be more planned and structured activities [38]. It is likely that individuals report activities when responding to self-report instruments that will not be detected as continuous ≥10 minutes by an accelerometer (e.g. playing intermittent sports, walking with stops to cross a road or to rest for some minutes). Thus, when using a stringent ≥10 minute criteria for fulfilling the recommendation, physical activity assessed by accelerometry may lead to an underestimation of the true prevalence.

Our data showed that the proportion fulfilling the recommended levels is highly dependent on whether MVPA is measured as bouted or accumulated non-bouted time; we observed a three-fold increase from 22% in bouted MVPA to 70% in accumulated non-bouted MVPA. Such patterns are also observed in previous studies from uniaxial accelerometry [10, 22, 39]. Moreover, when non-bouted MVPA is measured, our prevalence estimate is closer to the global estimate from self-reported physical activity [35], suggesting that such sporadic physical activity is also included in accelerometry when measuring non-bouted MVPA. Thus, understanding how different definitions of sufficiently active individuals may influence the prevalence estimates is important to inform public health recommendations and policies.

Furthermore, a recent meta-analysis showed maximal risk reduction in all-cause mortality at 24 minutes per day of accelerometry measured MVPA [40], which is similar to our chosen threshold for fulfilling the recommendations of 150 minute per week. The 24 minutes of MVPA for maximal risk reduction is also a substantially lower volume than what have previously been estimated from self-reported methods [41], indicating that the magnitude of the association between MVPA and mortality is in fact underestimated by self-reported methods. Accelerometry has been successfully implemented in surveillance systems and large cohorts [10, 22, 23, 42] and will likely be used in combination with self-reported physical activity in future large-scale studies. Thus, future studies that elucidates how different measurement tools influences the association with health outcomes is warranted.

Our prevalence estimates are similar to previous studies in Norwegian adults [14, 43], but higher than comparable estimates in Germany [42], Sweden [44], Portugal [10], the United States [11] and the United Kingdom [15, 22]. The observation of lower physical activity levels with higher age seems consistent across all studies measuring physical activity by accelerometry [10, 11, 14, 15, 22, 39, 42]. In previous studies, low levels of physical activity in older age are associated with disabilities such as difficulties in walking, pain and physical complaints [42, 45], indicating that the ageing process may influence physical activity levels. However, associations with disabilities disappear when controlling for morbidity confounders [45]. To date, there is no biological explanation for the consistent observed declines in physical activity levels with age, hence, encouraging older individuals to maintain or increase their physical activity levels may stimulate to healthy ageing and may thus have considerable impact on public health.

We found that men spent more time in accumulated non-bouted MVPA than women, whereas no sex differences were observed in bouted MVPA. In previous studies, male participants in studies from Norway [14, 43], the United States [11], Portugal [10], Germany [42] and the United Kingdom [15] accumulated more minutes of MVPA than female participants, whereas Swedish [39] and Chinese [13] women and men accumulated an equal amount of MVPA. The differences between the present study and the abovementioned studies may be due to different data processing protocols, thus, comparisons should be done with caution.

The inverse association between objectively assessed physical activity and BMI observed in the present study is consistent with previous studies [13, 14, 42]. Although a recent systematic review suggest that physical activity can prevent weight gain at the population level [46], methodological issues challenge this interpretation [47]. Basically, it is equally likely that lower levels of physical activity result in high BMI as *vice versa*, however, the direction in the association cannot be determined from cross-sectional designs [48].

Furthermore, our study demonstrated a positive association between bouted MVPA and educational level, which is consistent with studies from other high-income countries [13, 14, 49, 50]. Suggested reasons for lower MVPA in low education groups may include low perceived control, family responsibilities, poor perceived health, and financial and housing problems [51], as well as lack of knowledge of health benefits, attitudes and motivation towards physical activity [49]. Additionally, higher education is also associated with sedentary occupations [52], which may be compensated by an increased engagement in higher intensity leisure time physical activity [49]. In contrast, individuals with lower education are more likely to possess jobs including standing and/or walking, usually of light intensity physical activity [53, 54]. It is previously demonstrated that less sitting time at work may be associated with higher sitting time during leisure time [55]. Hence, those with lower education may be exposed to a more exhaustive working environment resulting in less leisure time physical activity of higher intensity due to the necessity of rest [53, 55, 56].

However, there were no differences in accumulated non-bouted MVPA between educational levels. As bouted MVPA may be planned and structured compared to non-bouted MVPA that may be more sporadic [38], this may also explain why non-bouted MVPA did not differ between educational levels: non-bouted MVPA may be performed during work hours to a larger extent in those with lower education as they may possess jobs including standing and sporadic walking that may reach accelerations corresponding to MVPA, which is in contrast to those with higher education that may have more sedentary occupations [52] and engage in more planned bouted MVPA during leisure time [52, 55].

Triaxial data resulted in more minutes of MVPA and less time spent sedentary than uniaxial data, which is consistent with previous studies in older women [8] and middle-aged adults [42]. Accordingly, the proportion meeting the current recommendations using uniaxial accelerometry data (18%) is approximately 20% lower compared with triaxial accelerometry data (22%). Moreover, this proportion is even larger when assessing non-bouted MVPA (triaxial: 70% vs. uniaxial: 55%). This corroborates previous observations suggesting triaxial accelerometry may capture more movement compared with uniaxial accelerometry [16], which may even be more pronounced in non-bouted MVPA compared with bouted MVPA.

In addition, our analyses suggested differences by sex and education levels when assessing uniaxial and triaxial accelerometry. When triaxial and uniaxial data are compared in laboratory settings, only small and typically non-significant differences are observed [18, 57]. This is possibly explained by the distinct activities performed in the laboratory studies, such as walking and running on a treadmill that have no unique medio-lateral and anterior-posterior accelerations in the hip, resulting in movements in the vertical axis being almost perfectly correlated with total 3-dimensial measurement of the similar movement, whereas behaviours during free-living conditions involve larger variation in movements, and thereby also more unique medio-lateral and anterior-posterior movements in the hip [18]. Additionally, this may explain why men accumulated more uniaxial CPM; as men may perform more walking and running than women, such differences may disappear when also analysing medio-lateral and anterior-posterior hip movements from triaxial accelerometry, which may be performed more by women. Nevertheless, the findings from the present study confirms earlier anticipations that triaxial accelerometry provide higher estimates of physical activity [16]. Thus, this illustrates that comparisons between different accelerometry processing methods should be done with caution and that tracking of physical activity across time is sensitive to accelerometry data collection and processing.

## Limitations

There are some limitations to this study. First, the intensity specific count-based cut-points in this study are based on laboratory studies using the relationship between acceleration and oxygen uptake during walking and running, which is then inter- or extrapolated to CPM for the respective intensities [29, 32]. Thus, the chosen cut-points are not calibrated to reflect the caloric intensity of activities that are biomechanically different from walking and running. For example, cycling at moderate intensity may be classified as light physical activity. However, according to the present study, triaxial accelerations seem to express a wider range of movements than uniaxial accelerations resulting in higher estimates of physical activity.

Further, this study included participants aged 40 years and older, whereas the validity studies for the intensity specific cut-points included participants with a mean age of ~25 years [29, 32]. As cardiorespiratory fitness decreases with increasing age [58–60], the employed cut-points in this study may be inappropriate for the older participants as the intensity specific thresholds are absolute. However, our study sample is suggested to represent the entire adult

population [24] and therefore, intensity-specific cut-points validated in young adults was considered the most appropriate.

A non-wear criteria of 20 minutes of consecutive 0 CPM seems to result in the lowest misclassification of wear and non-wear time [61]. However, this non-wear algorithm will exclude slightly more participants from final analyses compared with 60 minutes of consecutive 0 CPM [61]. The chosen algorithm for non-wear time in our study classified ~7 hours per day as non-wear time and only excluded 2.6% participants, in contrast to the study by Peeters et al. [61] where ~6% were excluded following the 20 minutes of consecutive 0 CPM algorithm. However, as no non-wear time algorithm is perfect, some misclassification of wear/non-wear time is inevitable within each trace of included participants. Considering the 24-hour protocol employed in the present study where 30% of the day was classified as non-wear time, it is likely that the method used may have removed too much true sedentary time which would inflate overall volume of activity estimates but not light physical activity and MVPA estimates directly. Moreover, our non-wear algorithm for excluding sleep has not been validated and may misclassify sedentary time.

The present study may be prone to accelerometer reactivity [62]. Some studies have observed higher physical activity levels on day one of recording compared with the following days [62], however, this is not consistent [63–66]. As it seems difficult to control for potential reactivity considering the need for information on the study´s purpose, potential reactivity is likely and has to be an acceptable limitation when employing accelerometry to measure individuals' daily physical activity levels and patterns.

Finally, selection bias may have affected our prevalence estimates [67]. A larger proportion of older participants and participants with lower education declined the invitation to wear an accelerometer. However, there were no differences in the distribution of smoking habits and BMI between those who declined and accepted the invitation. Moreover, the acceptance rate to the first visit in Tromsø 7 (65%), and especially the high acceptance rate for wearing the accelerometer (93% out of the 8346 attending the second visit) suggests a fair representativeness in the population. Additionally, the participants accepting to wear an accelerometer seem evenly distributed between educational levels (Table 1), suggesting an even distribution between social classes. Nevertheless, a non-respondent bias due to the most frail and unfit not participating cannot be ruled out.

## Strengths

This study included a large sample of adults and elderly, allowing us to assess the prevalence of physical activity in a large heterogeneous sample. Moreover, our population-based study can be considered to have a high acceptance rate (65%), with an even higher acceptance for wearing an accelerometer (93%). Finally, although no gold standard for measuring free living physical activity exists [68], we assessed the prevalence of physical activity using accelerometry, which is more accurate than self-reported methods when compared against the doubly labelled water technique [69, 70].

## Conclusion

The prevalence estimates of sufficiently active adults and elderly are more than three times higher (22% vs. 70%) when comparing triaxial bouted and non-bouted MVPA. Physical activity estimates are highly dependent on accelerometry data processing criteria and on ddifferent definitions of physical activity recommendations, which may influence prevalence estimates and tracking of physical activity patterns over time.

## Acknowledgments

The authors would like to acknowledge PhD Ola Løvsletten for advice on statistical analyses.

## Author Contributions

**Conceptualization:** Edvard H. Sagelv, Ulf Ekelund, Laila A. Hopstock, Bente Morseth.

**Data curation:** Edvard H. Sagelv, Ulf Ekelund, Sigurd Pedersen, Søren Brage, Alexander Horsch, Laila A. Hopstock, Bente Morseth.

**Formal analysis:** Edvard H. Sagelv, Ulf Ekelund, Søren Brage, Laila A. Hopstock, Bente Morseth.

**Funding acquisition:** Sameline Grimsgaard, Laila A. Hopstock, Bente Morseth.

**Investigation:** Edvard H. Sagelv, Søren Brage, Laila A. Hopstock, Bente Morseth.

**Methodology:** Edvard H. Sagelv, Ulf Ekelund, Laila A. Hopstock, Bente Morseth.

**Project administration:** Laila A. Hopstock, Bente Morseth.

**Resources:** Edvard H. Sagelv, Alexander Horsch, Bente Morseth.

**Software:** Edvard H. Sagelv, Alexander Horsch, Bente Morseth.

**Supervision:** Ulf Ekelund, Laila A. Hopstock, Bente Morseth.

**Validation:** Edvard H. Sagelv, Bente Morseth.

**Visualization:** Edvard H. Sagelv, Ulf Ekelund, Sigurd Pedersen, Bjørge H. Hansen, Sameline Grimsgaard, Bente Morseth.

**Writing – original draft:** Edvard H. Sagelv.

**Writing – review & editing:** Ulf Ekelund, Sigurd Pedersen, Søren Brage, Bjørge H. Hansen, Jonas Johansson, Sameline Grimsgaard, Anna Nordström, Alexander Horsch, Laila A. Hopstock, Bente Morseth.

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
