## [Decision Letter · Decision Letter 0]

4 Sep 2019

PONE-D-19-20213

Prevalence estimates of physical activity in adults and elderly from triaxial and uniaxial accelerometry. The Tromsø Study

PLOS ONE

Dear Mr Sagelv,

Thank you for submitting your manuscript to PLOS ONE. After careful consideration, we feel that it has merit but does not fully meet PLOS ONE’s publication criteria as it currently stands. Therefore, we invite you to submit a revised version of the manuscript that addresses the points raised during the review process.

We would appreciate receiving your revised manuscript by Oct 19 2019 11:59PM. To enhance the reproducibility of your results, we recommend that if applicable you deposit your laboratory protocols in protocols.io, where a protocol can be assigned its own identifier (DOI) such that it can be cited independently in the future. For instructions see: http://journals.plos.org/plosone/s/submission-guidelines#loc-laboratory-protocols

We look forward to receiving your revised manuscript.

Kind regards,

Fernando C. Wehrmeister

Academic Editor

PLOS ONE

Journal Requirements:

3. Please include your tables as part of your main manuscript and remove the individual files. Please note that supplementary tables (should remain/ be uploaded) as separate "supporting information" files

Additional Editor Comments (if provided):

The manuscript is very interesting and important for the field. Based on the reviewers' comments and by my assessment of the paper, I feel that this manuscript has a great potential to be published, since the authors can handle comments below. I would like to point out one more issue that was not raised by the reviewers: the choice of statistical tests. I strongly suggest to authors to choose between heterogeneity or trend p-values, not present both in tables. Maybe you can test the depart from linearity to make your choice. Also, more details about the methods of the Tromsø study should be provided in the methods section.

Reviewers' comments:

Reviewer's Responses to Questions

**Comments to the Author**

1. Is the manuscript technically sound, and do the data support the conclusions?

Reviewer #1: Partly

Reviewer #2: Partly

2. Has the statistical analysis been performed appropriately and rigorously? 

Reviewer #1: No

Reviewer #2: Yes

3. Have the authors made all data underlying the findings in their manuscript fully available?

Reviewer #1: Yes

Reviewer #2: Yes

4. Is the manuscript presented in an intelligible fashion and written in standard English?

Reviewer #1: Yes

Reviewer #2: Yes

5. Review Comments to the Author

Reviewer #1: Congratulations for your work. The article is very interesting, but some changes are necessary.

The main problem is the use of prevalence accordingo to WHO recommendation instead averages. This is a problem with accelerometry because recommendations were built based on questionaires and not in a mechanical measure. Therefore prevalence obtained with accelerometer are not applicable to WHO recomendations I strongly suggest rephrasing article for averaging measures.

Introdution:

Line 59-60:

Questionnaires are still very important because accelerometry does not include information such as domains of physical activity or subjective characteristics. Therefore, the term "replace" used here needs to be changed

Line 77-79:

Please confirm in the text if the limit used is validated for one or three axys

Methods:

It is unclear the year of data collect.

Line 161: Was used an algorithm to sleep detection or only visual inspection based on lower movimentation period? Please specify this topic and if necessary, include in limitations of study.

Line 181: Again, the main problem with the study is the use of prevalence rather than averages. Please use averages / medians in the analysis

Confirm that the MVPA variable is symmetrical to present the means. If not, use nonparametric tests and present medians.

Results:

Figure 1: I suggest exclude this Figure and thinking of other alternatives such as % of individuals with 0min of MVPA. This Figure may result in incorrect information about precentage of individuals meeting WHO recomendations because it is based on accelerometry and is not comparable with data from questionaries.

Discussion:

Please, discuss what is the meaning of MVPA with and without bouts and which movement patterns they can represent.

Lines 377-380: I suggest rephrasing the sentence. This frase seems blame older people for the lower levels of physical activity compared to Young people. Other mechanisms, such as, worse sleep indicators, lower muscle strengh and health problems may explain these changes across the life. Please, review the sentence.

Lines 382-388:

Again, these results may be due to differnece in physical activity contructs based on different intensities and bout criteria. The discussion needs further explanation about this.

Please, add discussion about mechanisms between physical activity and IMC. Even with reverse causality, some suggestion for the findings is necessary.

Please, add to the discussion a topic about what to do with the article results. Can the affect measurements in longitudinal studies using uni and triaxials accelerometers across the time? What are the authors recommendations based on article findings?

Line470-472: High acceptance rate does not mean lack of bias. You need to compare sample features and losses. At least, suggest in which direction high representativeness in the population and among different educational levels may affect the findings presented.

After the limitations, please report the strengths of the study.

Conclusion

The conclusion needs reformulation according to the suggestions of the article.

Reviewer #2: This is a cross-sectional study, which has investigated the prevalence of accelerometer-based physical activity in mid-age adults from the Tromso Study in Norway. Moreover, the authors compared estimates derived by triaxial and uniaxial accelerometers. Currently, there is substantial interest and need for population estimates of physical activity based on objective measures. The context of this paper concerning the existing literature is well described, the study is well designed, and the conclusions are mostly aligned with the methods and findings. I have minor comments that are offered for the authors to consider to clarity the paper.

1- Sorry, but I could not see tables 3, 4 and 5. They were not provided in the main manuscript file, and I could not find any other alternative file with the submission.

2- Would be essential to report on the abstract the place where the accelerometers were worn.

3- It is not clear in the abstract whether the prevalence of physical activity (PA) presented (22% and 70%) are from triaxial or uniaxial. Given the comparison between triaxial and uniaxial is part of the study aims, it would be important to make it clear in the abstract.

4- A significant amount of the study relied on comparing the compliance with physical activity guidelines. However, it is essential to note that the current guidelines of 150 minutes per week of MVPA are mostly based on shreds of evidence from self-report data. The manuscript would gain by discussing this issue.

5- The sampling process and the design of the study are not clear. Is the Tromso 7 a follow-up measure from Tromso 1, or is it a new cohort that started in 2015? Is this a cohort study or a series of cross-sectional studies?

6- Please consider describing the sample from the starting sample size at recruitment, and the comparison of the analytical sample with the original sample. How to the 5918 participants compare with the overall population? The strength is the population-based design, but the reader cannot ascertain how representative the sample is.

7- Page 8, line 187 – see ‘triacial’

8- Presenting standard deviations and means would be preferable instead of standard errors for the means, as it gives to the reader a clearer idea on the variability of the data.

9- Page 9, line 213-215. It is not clear. Please review.

10- Although the difference in the prevalence of PA 10-min bouts between the triaxial and uniaxial was 4 percentage points, there was a 15-percentage points difference when non-bout was considered. Could it be further discussed in the discussion section?

11- Page 14, line 353; please review ‘These’ and use ‘those’ instead.

12- Page 17, line 410. Please note that for a prevalence of 22%, ‘4% lower’ would mean 21.2%. The difference between 22% and 18% represents 4 percentage points…Alternatively, the authors could say that the prevalence of 18% is approximately 20% lower than prevalence of 22%.

13- The conclusion retells the results and does not provide directions. The ‘so what?’ is missing in conclusion.

14- English language wording needs to be checked, as there are minor errors throughout the paper.

6. PLOS authors have the option to publish the peer review history of their article (what does this mean?). If published, this will include your full peer review and any attached files.

Reviewer #1: No

Reviewer #2: No

---

## [Author Response · Author response to Decision Letter 0]

27 Sep 2019

Review Comments to the Author

Reviewer #1: Congratulations for your work. The article is very interesting, but some changes are necessary.

The main problem is the use of prevalence accordingo to WHO recommendation instead averages. This is a problem with accelerometry because recommendations were built based on questionaires and not in a mechanical measure. Therefore prevalence obtained with accelerometer are not applicable to WHO recomendations I strongly suggest rephrasing article for averaging measures.

Response: Thank you for your thorough work reviewing our manuscript. Below is our response to your comments.

The reviewer mentions an important concern. However, we respectfully disagree regarding the prevalence estimates. Although WHO recommendations are based on self-report, we still think that presenting prevalences estimates by accelerometry is of interest and consistent with previous studies (our reference nr 10, 22, 23), and also a previous study published in this journal (reference nr 42). In addition, a recent meta-analysis shows that the maximal risk reduction for all-cause mortality was observed at 24 min per day of accelerometry measured MVPA (Ekelund et al., 2019, BMJ, now our reference nr 43), similar to the 150 minute per week thresholds we used when estimating prevalence of inactivity. This is also considerably lower than previous estimates from self-report (Stamatakis et al., 2019, J Am College Cardiol, now our reference nr 44), suggesting that the magnitude of associations with mortality based on self-report are underestimated. Thus, we consider reporting prevalence estimates from accelerometry is justified, emphasizing the importance and complexity of the measurement methods when estimating prevalences.

Reviewer 2 also comment on this issue, suggesting to include a paragraph addressing this issue in the discussion. Accordingly, this is done (L414-424). We hope our attempt in addressing this issue is satisfactory.

Introdution:

Line 59-60:

Questionnaires are still very important because accelerometry does not include information such as domains of physical activity or subjective characteristics. Therefore, the term "replace" used here needs to be changed

Response: We fully agree with the reviewer and accordingly, “replace” is removed from the sentence.

Line 77-79:

Please confirm in the text if the limit used is validated for one or three axys

Response: Unfortunately, we do not understand the comment from the reviewer. Could the reviewer please clarify this comment, and we hope we can get the chance to address this comment in a second round of revisions. 

Methods:

It is unclear the year of data collect.

Response: We apologize for this not being explicitly clear. Accordingly, we have stated the following in line 105: “The present study includes participants from the seventh survey conducted in 2015-16.”

Line 161: Was used an algorithm to sleep detection or only visual inspection based on lower movimentation period? Please specify this topic and if necessary, include in limitations of study.

Response: Besides the non-wear time algorithm, there were no additional algorithm to exclude sleep, as it seems that the non-wear time algorithm also excluded sleep. We agree that this is a limitation and accordingly, we have included this under limitation (please see L533-534).

Line 181: Again, the main problem with the study is the use of prevalence rather than averages. Please use averages / medians in the analysis

Response: Please see our answer above.

Confirm that the MVPA variable is symmetrical to present the means. If not, use nonparametric tests and present medians.

Response: The Kolmogorov-Smirnov (the Shapiro-Wilk test is unavailable to sample sizes >50 in SPSS version 25) comes out as significant, indicating that the MVPA variable deviates from normal distribution and is positively skewed. However, as presented under statistical analysis, from visual inspection of residuals in the ANOVA and ANCOVA, the data can be considered to follow normal distribution. This has been confirmed by our statistician, suggesting that parametric analyses are appropriate. 

Results:

Figure 1: I suggest exclude this Figure and thinking of other alternatives such as % of individuals with 0min of MVPA. This Figure may result in incorrect information about precentage of individuals meeting WHO recomendations because it is based on accelerometry and is not comparable with data from questionaries.

Response: We respectfully disagree, please see our response as discussed above. 

Discussion:

Please, discuss what is the meaning of MVPA with and without bouts and which movement patterns they can represent.

Response: We have presented the meaning in the introduction, please see L84-90.

As the WHO recommendations states bouted MVPA while the updated American guidelines do not require bouts of 10 minutes, comparing bouts and no bouts have, in our view, applicability. The updated UK guidelines have now also removed the 10-min bout criteria two weeks ago (https://www.gov.uk/government/publications/physical-activity-guidelines-uk-chief-medical-officers-report). Finally, the WHO guidelines are under revision, which make our presentation of bouted and non-bouted MVPA relevant, as they clearly show the difference between these two approaches. 

However, what we have not mentioned is what types of movement they may represent. Unfortunately, what movement they may represent are difficult to estimate, as accelerometry only measures acceleration, and in our case, acceleration in the hip. Whether non-bouted MVPA represents jumping, walking, running, sprinting, or house work is unknown. The only interpretation possible with this data, is to suggest that <10 min MVPA may be more sporadic and may represent more impulsive movements, while >min MVPA may be more planned/or intended as they are continuous. Accordingly, we have implemented a sentence in this regard (see L400-412). Thank you for this suggestion.

Lines 377-380: I suggest rephrasing the sentence. This frase seems blame older people for the lower levels of physical activity compared to Young people. Other mechanisms, such as, worse sleep indicators, lower muscle strengh and health problems may explain these changes across the life. Please, review the sentence.

Response: As stated in this paragraph, lower levels of PA are associated with disabilities in walking, pain etc., however, this association disappeared when controlling for morbidity. Thus, implying a causal link between worse health and low PA levels and not vice versa, should be interpreted with caution. As we have stated, no biological explanation exists for the decline in PA levels, indicating that perhaps this association is the other way around: lower PA levels results in worse sleep, lower strength (indeed, no stimuli of muscular movements will result in low muscle strength) and poor health outcome (there are multiple studies showing that those who exercise have higher cardiorespiratory fitness in older age (Wilson and Tanaka, 2000, Circulation, Tanaka and Seals, 2008, J Physiol). Nevertheless, we have changed the phrasing in order to avoid blaming older individuals.

Lines 382-388:

Again, these results may be due to differnece in physical activity contructs based on different intensities and bout criteria. The discussion needs further explanation about this.

Response: Thank you for addressing this. In this paragraph, we have stated that comparisons in this paragraph should be made with caution due to different processing, please see L439-445. We hope this is satisfactory for addressing this important issue.

Please, add discussion about mechanisms between physical activity and IMC. Even with reverse causality, some suggestion for the findings is necessary.

Response: Thank you for your suggestion. We have revised the text to address this issue, please see L447-452. 

Please, add to the discussion a topic about what to do with the article results. Can the affect measurements in longitudinal studies using uni and triaxials accelerometers across the time? What are the authors recommendations based on article findings?

 Response: Thank you for this excellent suggestion. We have amended the discussion as suggested, please see L500-502.

Line470-472: High acceptance rate does not mean lack of bias. You need to compare sample features and losses. At least, suggest in which direction high representativeness in the population and among different educational levels may affect the findings presented.

Response: Thank you for this important comment we fully agree that lack of bias may never be removed. We have tried to discuss this issue (please see L543-552). Additionally, we have compared the total sample with the accelerometer sample in age, weight, height and BMI (L241-275).

After the limitations, please report the strengths of the study.

Response: Thank you for this comment, accordingly, this is done (L555-561). 

Conclusion

The conclusion needs reformulation according to the suggestions of the article.

Response: Thank you for the suggestion. Accordingly, we have tried to revise the conclusion to reflect the aims and content to a larger degree, please see L564-568. 

Final note:

Thank you for your thorough work reviewing our paper. We think your comments and suggestions to revision have improved our paper. 

Reviewer #2: This is a cross-sectional study, which has investigated the prevalence of accelerometer-based physical activity in mid-age adults from the Tromso Study in Norway. Moreover, the authors compared estimates derived by triaxial and uniaxial accelerometers. Currently, there is substantial interest and need for population estimates of physical activity based on objective measures. The context of this paper concerning the existing literature is well described, the study is well designed, and the conclusions are mostly aligned with the methods and findings. I have minor comments that are offered for the authors to consider to clarity the paper.

Response: Thank you for your thorough work reviewing our manuscript. Below is our response to your comments. 

1- Sorry, but I could not see tables 3, 4 and 5. They were not provided in the main manuscript file, and I could not find any other alternative file with the submission.

Response: We are sorry to read that you have not have any chance to see the tables. These tables were large and thus uploaded as additional files. We have now included them in the manuscript. 

2- Would be essential to report on the abstract the place where the accelerometers were worn.

Response: Thank you for the suggestion, this is now included in the abstract (L38). 

3- It is not clear in the abstract whether the prevalence of physical activity (PA) presented (22% and 70%) are from triaxial or uniaxial. Given the comparison between triaxial and uniaxial is part of the study aims, it would be important to make it clear in the abstract.

Response: Thank you for the suggestion, this is now included (L42). 

4- A significant amount of the study relied on comparing the compliance with physical activity guidelines. However, it is essential to note that the current guidelines of 150 minutes per week of MVPA are mostly based on shreds of evidence from self-report data. The manuscript would gain by discussing this issue.

Response: This issue is important to address. Accordingly, we have included a paragraph addressing this very issue (please see L413-423).

5- The sampling process and the design of the study are not clear. Is the Tromso 7 a follow-up measure from Tromso 1, or is it a new cohort that started in 2015? Is this a cohort study or a series of cross-sectional studies?

Response: We apologize for this being unclear. The Tromsø Study is an ongoing cohort study, inviting previous participants as well as random samples in repeated surveys named Tromsø 1, Tromsø 2, Tromsø 3, Tromsø 4, Tromsø 5, Tromsø 6 and Tromsø 7 (last survey so far). Each survey therefore represents a combination of longitudinal and cross-sectional cohorts. In the current analyses we used a sample from the seventh survey, Tromsø 7, as this was the first survey that measured PA by accelerometry in a larger sample, limiting this study to a cross-sectional design. 

6- Please consider describing the sample from the starting sample size at recruitment, and the comparison of the analytical sample with the original sample. How to the 5918 participants compare with the overall population? The strength is the population-based design, but the reader cannot ascertain how representative the sample is.

Response: Thank you for the comment. We agree that representativeness is an important issue and have therefore included a new table (now table 1) showing the descriptive characteristics of the original sample as well, which is compared to the sub-sample in terms of age, BMI, weight and height.

7- Page 8, line 187 – see ‘triacial’

Response: Thank you for your comment, this is now corrected.

8- Presenting standard deviations and means would be preferable instead of standard errors for the means, as it gives to the reader a clearer idea on the variability of the data.

Response: we have substituted SEM with SD, as suggested.

9- Page 9, line 213-215. It is not clear. Please review.

Response: Ok, we have now tried to make it clearer, please see L212-216.

10- Although the difference in the prevalence of PA 10-min bouts between the triaxial and uniaxial was 4 percentage points, there was a 15-percentage points difference when non-bout was considered. Could it be further discussed in the discussion section?

Response: We have briefly discussed this in the revised version of the manuscript (L476-483).

11- Page 14, line 353; please review ‘These’ and use ‘those’ instead.

Response: Thank you for your comment, this is now corrected.

12- Page 17, line 410. Please note that for a prevalence of 22%, ‘4% lower’ would mean 21.2%. The difference between 22% and 18% represents 4 percentage points…Alternatively, the authors could say that the prevalence of 18% is approximately 20% lower than prevalence of 22%.

Response: Thank you for this comment. This adds an interesting point of presenting and interpretation. Please see the revised sentence (L476-483).

13- The conclusion retells the results and does not provide directions. The ‘so what?’ is missing in conclusion.

Response: Thank you, we have accordingly revised the conclusion to better reflect the aims and content, which may hopefully to a larger degree provide some implications of the study (L563-567). 

14- English language wording needs to be checked, as there are minor errors throughout the paper.

Response: Thank you for this comment. We have now checked spelling and grammar, hopefully the manuscript reads better now.

---

## [Decision Letter · Decision Letter 1]

29 Oct 2019

PONE-D-19-20213R1

Prevalence estimates of physical activity in adults and elderly from triaxial and uniaxial accelerometry. The Tromsø Study

PLOS ONE

Dear Mr Sagelv,

Thank you for submitting your manuscript to PLOS ONE. After careful consideration, we feel that it has merit but does not fully meet PLOS ONE’s publication criteria as it currently stands. Therefore, we invite you to submit a revised version of the manuscript that addresses the points raised during the review process.

We would appreciate receiving your revised manuscript by Dec 13 2019 11:59PM. To enhance the reproducibility of your results, we recommend that if applicable you deposit your laboratory protocols in protocols.io, where a protocol can be assigned its own identifier (DOI) such that it can be cited independently in the future. For instructions see: http://journals.plos.org/plosone/s/submission-guidelines#loc-laboratory-protocols

We look forward to receiving your revised manuscript.

Kind regards,

Fernando C. Wehrmeister

Academic Editor

PLOS ONE

Additional Editor Comments (if provided):

Congratulations for the authors for this new version of the paper. Based on the assessment of the reviewers and myself, I believe that the paper deserves be published in Plos One. However, some minor points need attention. According to reviewer 2, the paper will benefit from a more detailed discussion on the impact of these kind of measure on public health. Authors should consider expand the discussion on this specific point. Other comment is regarding the title. I will suggest that the authors use terms like "levels" instead prevalence. Why? The paper has 6 tables showing details of the parameters obtained with accelerometer, and only one figure for prevalence. So the component of "levels" in the manuscript is crucial for understanding it.

Reviewers' comments:

Reviewer's Responses to Questions

**Comments to the Author**

1. If the authors have adequately addressed your comments raised in a previous round of review and you feel that this manuscript is now acceptable for publication, you may indicate that here to bypass the “Comments to the Author” section, enter your conflict of interest statement in the “Confidential to Editor” section, and submit your "Accept" recommendation.

Reviewer #1: (No Response)

Reviewer #2: All comments have been addressed

2. Is the manuscript technically sound, and do the data support the conclusions?

Reviewer #1: Partly

Reviewer #2: Yes

3. Has the statistical analysis been performed appropriately and rigorously? 

Reviewer #1: Yes

Reviewer #2: Yes

4. Have the authors made all data underlying the findings in their manuscript fully available?

Reviewer #1: Yes

Reviewer #2: Yes

5. Is the manuscript presented in an intelligible fashion and written in standard English?

Reviewer #1: Yes

Reviewer #2: Yes

6. Review Comments to the Author

Reviewer #1: I understand your argument when use the reference “Ekelund et al., 2019, BMJ” to justify the use of 150 minutes. In other hand, accelerometry measure includes many decisions not included in self-report. When asked about your habitual physical activity, the answer probably will not include an exact definition of bout or intensity. Thus, if an individual practice sports for 1h, then this 1h will not necessarily be classified as MVPA time in accelerometry (including bout criteria, this will be very different). Therefore, to reach recommendations with accelerometry definitions is very difficult resulting in lower percentages. In addition, prevalence information, in many cases is used to show a health scenario in the world, or in the local context. If you want to keep this kind of measure, I suggest improving the discussion on the differences between prevalence measure by method and how this is used in public health. This measure may be mistakenly used as “the true” about health/physical activity scenario of population and severity of this scenario may vary greatly according to accelerometry decisions. The message of the article is unclear about what to do with this prevalence and how it may impact or apply to public health. Often, “more physical activity” is the final message, but in your case, you show a percentage of population reaching an expected value, so what to do with this percentage so different from previous self-report studies needs further explanations.

Reviewer #2: Thank you for the opportunity to review this manuscript. I do not have further considerations.

7. PLOS authors have the option to publish the peer review history of their article (what does this mean?). If published, this will include your full peer review and any attached files.

Reviewer #1: No

Reviewer #2: No

---

## [Author Response · Author response to Decision Letter 1]

6 Nov 2019

Response to the editor and reviewers

Additional Editor Comments (if provided):

Congratulations for the authors for this new version of the paper. Based on the assessment of the reviewers and myself, I believe that the paper deserves be published in Plos One. However, some minor points need attention. According to reviewer 2, the paper will benefit from a more detailed discussion on the impact of these kind of measure on public health. Authors should consider expand the discussion on this specific point. Other comment is regarding the title. I will suggest that the authors use terms like "levels" instead prevalence. Why? The paper has 6 tables showing details of the parameters obtained with accelerometer, and only one figure for prevalence. So the component of "levels" in the manuscript is crucial for understanding it.

Answer: Thank you for the opportunity for this second revisions to further improve our paper. According to your suggestions, we have further expanded our discussion on the interpretation of our prevalence estimates and how this impacts public health (please see L398-440). We agree on the title change and we have accordingly changed our title to “Physical activity levels in adults and elderly from triaxial and uniaxial accelerometry. The Tromsø Study”. We have also included “levels” as a part of our aim: Abstract, see L33 and main text: see L92.

6. Review Comments to the Author

Reviewer #1: I understand your argument when use the reference “Ekelund et al., 2019, BMJ” to justify the use of 150 minutes. In other hand, accelerometry measure includes many decisions not included in self-report. When asked about your habitual physical activity, the answer probably will not include an exact definition of bout or intensity. Thus, if an individual practice sports for 1h, then this 1h will not necessarily be classified as MVPA time in accelerometry (including bout criteria, this will be very different). Therefore, to reach recommendations with accelerometry definitions is very difficult resulting in lower percentages. In addition, prevalence information, in many cases is used to show a health scenario in the world, or in the local context. If you want to keep this kind of measure, I suggest improving the discussion on the differences between prevalence measure by method and how this is used in public health. This measure may be mistakenly used as “the true” about health/physical activity scenario of population and severity of this scenario may vary greatly according to accelerometry decisions. The message of the article is unclear about what to do with this prevalence and how it may impact or apply to public health. Often, “more physical activity” is the final message, but in your case, you show a percentage of population reaching an expected value, so what to do with this percentage so different from previous self-report studies needs further explanations.

Answer: Thank you for the suggestion. We agree on the importance of this and we have accordingly further expanded our discussion on the interpretation of our prevalence estimates and how this impacts public health (please see L398-440).

Reviewer #2: Thank you for the opportunity to review this manuscript. I do not have further considerations.

Answer: Thank you for reviewing our paper.

---

## [Editor Report · Decision Letter 2]

11 Nov 2019

Physical activity levels in adults and elderly from triaxial and uniaxial accelerometry. The Tromsø Study

PONE-D-19-20213R2

Dear Dr. Sagelv,

We are pleased to inform you that your manuscript has been judged scientifically suitable for publication and will be formally accepted for publication once it complies with all outstanding technical requirements.

With kind regards,

Fernando C. Wehrmeister

Academic Editor

PLOS ONE

Additional Editor Comments (optional):

The authors appropriate deal with the comments from the editor and the reviewer. The paper is very nice and will be important for the field of physical activity objectively measured. Congratulations!
---

## [Editor Report · Acceptance letter]

18 Nov 2019

PONE-D-19-20213R2 

Physical activity levels in adults and elderly from triaxial and uniaxial accelerometry. The Tromsø Study 

Dear Dr. Sagelv:

I am pleased to inform you that your manuscript has been deemed suitable for publication in PLOS ONE. Congratulations! Your manuscript is now with our production department. 

With kind regards,

on behalf of

Dr. Fernando C. Wehrmeister 

Academic Editor

PLOS ONE